

# Assessment of impacts of agricultural and climate change scenarios on watershed water quantity and quality, and crop production

Awoke D. Teshager[1], Philip W. Gassman[2], Justin T. Schoof[3], and Silvia Secchi[3]

[1]Environmental Resources and Policy, Southern Illinois University Carbondale, 400 N Westridge Dr, Carbondale IL 62901, USA
[2]Department of Economics, Center for Agricultural and Rural Development, Iowa State University, 560A Heady Hall, Ames IA 50011, USA
[3]Geography and Environmental Resources, Southern Illinois University Carbondale; Faner Hall, Carbondale IL 62901, USA

*Correspondence to*: Awoke D. Teshager (dagnew.awoke@siu.edu)

**Abstract.** Modeling impacts of agricultural scenarios and climate change on surface water quantity and quality provides useful information for planning effective water, environmental, and land use policies.
Despite the significant impacts of agriculture on water quantity and quality, limited literature exists that describes the combined impacts of agricultural land use change and climate change on future bioenergy crop yields and watershed hydrology. In this study, the Soil and Water Assessment Tool (SWAT) eco-hydrological model was used to model the combined impacts of five agricultural land use change scenarios and three downscaled climate pathways (representative concentration pathways, RCPs) that were created from an
ensemble of eight atmosphere-ocean general circulation models (AOGCMs). These scenarios were implemented in a well calibrated SWAT model for the Raccoon River watershed (RRW) located in western Iowa. The scenarios were executed for the historical baseline, early-century, mid-century, and late-century periods. The results indicate that historical and more corn intensive agricultural scenarios with higher $CO_2$ emissions consistently result in more water in the streams and greater water quality problems, especially late
in the 21$^{st}$ century. Planting more switchgrass, on the other hand, results in less water in the streams and water quality improvements relative to the baseline. For all given agricultural landscapes simulated, all flow, sediment and nutrient outputs increase from early-to-late century periods for the RCP4.5 and RCP8.5 climate scenarios. We also find that corn and switchgrass yields are negatively impacted under RCP4.5 and RCP8.5 scenarios in the mid and late 21$^{st}$ century.



## 1   Introduction

Land use change and climate change are at the forefront of various pressures that are expected to alter 21st century land ecosystems (Ostberg et al., 2015; Heffernan et al., 2014; Howells et al., 2013; Moore et al., 2012). Both factors have been shown to independently or collectively greatly impact watershed

hydrology and/or water quality across a tremendous range of scales, as shown in literally hundreds of studies in the existing literature (e.g., Wilson and Weng, 2011; Jha et al., 2006, 2010; Secchi et al., 2011; Panagopoulos et al., 2015; Tan et al., 2015; Mehdi et al., 2015a, 2015b)  These land use and climate change impacts pose potentially serious issues for specific communities (Kundzewicz et al., 2007) and also for large regions or whole countries (Heffernan et al., 2014; Howells et al., 2013; Moore et al., 2012). Thus, it

is urgent to evaluate the potential impacts of combined future land use and climate change on different ecosystems and hence planning effective water, environmental, and land use policies (Heffernan et al., 2014).

Key agricultural production regions are critical ecosystems that may be adversely impacted by future land use change and climate change (Moore et al., 2012; Howells et al., 2013). An important component of likely

future agricultural land use change is the increased development of biofuel cropping systems, which are projected to require 37 million ha by the year 2030 (Howells et al., 2013). Extensive expansion of the biofuel industry has occurred in the U.S. Corn Belt region, primarily in the form of corn grain-based ethanol (RFA, 2011). Several studies report the potential of increased water quality problems or other ecosystem degradation due to the expansion of corn production in the Corn Belt region (e.g., Donner and Kucharik, 2008; Simpson et

al., 2008; Jha et al., 2010; Secchi et al., 2011; Wright and Wimberly, 2013). These potential problems underscore the need to investigate the environmental impacts of more widespread adoption of advanced perennial biofuel crops such as switchgrass, which has been found to provide multiple environmental benefits including carbon sequestration, soil water nutrient scavenging, remediating contaminated soil and/or providing suitable habitat for grassland birds (Khanna et al., 2008, Secchi et al., 2008; Vadas, 2008;

Keshwani and Cheng, 2009).  Schmer et al. (2008) investigated the net energy of cellulosic ethanol made from switchgrass over a five-year time period and found that switchgrass ethanol production resulted in 540% more renewable than nonrenewable energy consumed and 94% less GHG emissions than gasoline production. Vadas et al. (2008) further suggested that switchgrass may be best suited in highly erodible lands, considering its environmental benefits, in investigating economics and energy of ethanol production

from alfalfa, corn and switchgrass. Moreover, various researchers have shown the benefit of switchgrass in reducing sediment and nutrient yields from cropland landscapes (e.g., Schilling et al., 2008; Wu et al., 2012; Zhou et al., 2015).

A variety of tools have been developed that can be used to investigate the impacts of climate change and/or land use change in agricultural ecosystems including the Soil and Water Assessment Tool (SWAT)



ecohydrological model (Arnold et al., 1998; 2012; Williams et al., 2008). SWAT has been used worldwide to investigate an extensive array of hydrological and/or pollutant transport problems across a wide range of watershed scales (Gassman et al., 2007; 2014b; Krysanova and White, 2015; Bressiani et al., 2015; Gassman and Wang, 2015). An extensive review of earlier SWAT literature revealed that applications of

the model for climate change and land use scenarios were two of the key application trends occurring at that time (Gassman et al., 2007). More recent reviews of SWAT literature confirm that this trend has continued unabated (Krysanova and White, 2015; Gassman et al., 2014a) and current documentation of the SWAT literature indicates that roughly five hundred studies describe some type of climate change application while over three hundred studies report the effects of land use change (CARD, 2016).

10       An emerging trend in this overall subset of SWAT literature is the application of the model for combined climate change and land use change impacts (Krysanova and White, 2015; Gassman et al., 2014a); over seventy combined impact studies have now been documented (CARD, 2016). Such studies first were reported for Chinese conditions (Li al., 2004) which now include applications focused on capturing the effects of historical land use change due to the influence of Chinese government programs

(Zuo et al., 2016; Liu et al., 2015; Liu et al., 2013) and scenarios that reflect hypothetical shifts between various percentages of urban, forest, agricultural and other land use (Zhang et al., 2016; 2015; Wu et al., 2015). Similar types of combined SWAT climate change/land use change studies have been performed in other regions including Asia (Sayasane et al., 2015; Singkran et al., 2015; Tan et al., 2015), Europe (Serpa et al., 2015; Mehdi et al., 2015b; Guse et al., 2015) and North America (Mehdi et al., 2015a; Neupane and

Kumar, 2015; Goldstein and Tarhule, 2015).

      Several SWAT studies have focused specifically on the combined impacts of climate change and land use change on hydrological and/or pollutant responses within an agricultural context. Mehdi et al. (2015a; 2015b) describe similar methodologies of analyzing future agricultural land use and management scenarios for forecasted land use for watersheds that drain portions of Quebec and Vermont or an area in the

Bavarian region of Germany, respectively, in conjunction with projected future climate change. Guse et al. (2015) discuss the impacts of three land use scenarios, which represent shifts in cropping and grassland allocations, in combination with a RCM projection on future macroinvertebrate and fish habitat for a watershed in northern Germany. Neupane and Kumar (2015) report the impacts of expanded corn production within projected late 21st century climate conditions for a watershed in eastern South Dakota.

Other studies (Wu et al., 2013; Hoque et al., 2014; Goldstein and Tarhule, 2015) describe the impacts of introducing perennial bioenergy crops within cropland landscapes for varying predicted future climate conditions for watersheds located in the U.S. Corn Belt or Great Plains regions. Collectively, these studies reveal that hydrologic and pollutant transport characteristics for cropland landscapes can be very sensitive to shifts in land use and/or climate.



A complex set of factors drives cropping system decisions for a given Corn Belt region land parcel including crop prices, land productivity, previous years' profits, costs for fertilizer, energy, pesticides and other inputs, neighbors' choices, government programs and available markets for supporting production of a specific crop. Future development of infrastructure would need to occur to support perennial bioenergy

crop production in the Corn Belt region. In contrast, three cellulosic ethanol plants are being developed or in operation in the Corn Belt region that rely on corn stover (Peplow, 2014; ENERGY.GOV, 2015), a trend that could drive even more demand for corn production. Thus, Additional research is needed to ascertain the hydrologic and water quality impacts of possible increased corn production versus perennial biofuel crop adoption within projected future climate conditions for Corn Belt region stream systems.

Thus the focus of this study is to investigate the combined hydrologic and water quality impacts of potential future bioenergy crop production and projected future climate change for cropland landscapes of the Raccoon River watershed (RRW) located in western Iowa. The RRW is characterized by intensive row crop agriculture dominated by corn and soybean production, widespread use of subsurface tile drainage systems within flatter cropland landscapes and intensive nitrogen and phosphorus inputs in the form of

inorganic fertilizers and livestock manure. The Des Moines Water Works (DMWW), the largest such system in Iowa, relies on the Raccoon River as a key source of drinking water for Des Moines metropolitan area. The DMWW was forced to build what is believed to be the world's largest nitrate removal facility in 1991 in order to meet U.S. federal drinking water standards (White, 1996; DMWW, 2015) and operated the facility a record-breaking 111 days in 2015. The DMWW also filed a law suit against three upstream Iowa

counties in the watershed for their excessive nitrate load to the Raccoon River.

Several previous studies have been conducted for the RRW stream system with SWAT to investigate the hydrologic and water quality impacts of alternative cropping systems including systems consisting solely of perennial grasses such as switchgrass and/or the inclusion of alfalfa in rotation with row crops (Schilling et al., 2008; Jha et al., 2010; Gassman et al., 2015). Jha and Gassman (2014) further investigated

the impacts of potential future climate change on RRW hydrology using an ensemble of 10 atmosphere-ocean general circulation models (AOGCMs) and typical cropping systems consisting of rotations of corn and soybean. However, analysis of the combined effects of agricultural land use change and climate change are currently lacking for the RRW and for the Corn Belt region in general, especially in the context of evaluating the impacts of potential biofuel cropping systems. To address this gap, a SWAT analysis is

performed in this study for the RRW that incorporates five agricultural scenarios, three 21st century future climate periods (early, mid and late), and three greenhouse gas (GHG) emission pathways (RCP2.6, RCP4.5 and RCP8.5) that were represented within an ensemble of eight AOGCMs that were included in Phase 5 of the Coupled Model Intercomparison Project (CMIP5) (Taylor et al., 2012). The analysis is performed using an improved RRW SWAT model (Teshager et al., 2015) that allows analysis of typical



row crop and/or perennial biofuel cropping systems at a refined spatial scale representative of field-level land parcels. Thus, the objectives of this study are to: (1) describe the methodology used to develop the combined agricultural land use change and future climate change projections, and (2) quantify the effects of the combined scenarios on future RRW hydrology, water quality and crop yields.

## 2  Study Area

The RRW drains a total area of 9393 km$^2$ from portions of 17 counties in western central Iowa (Fig. 1). The RRW is also composed of two 8-digit watersheds as defined by the U.S. federal watershed classification system (USGS, 2013) which are referred to as the North Raccoon and South Raccoon watersheds. The North Raccoon watershed is dominated by flat land and poor surface drainage while the

South Raccoon watershed is characterized by higher slopes, steeply rolling hills, and well developed drainage (Agren, 2011). Fertilizer and livestock manure applications on cropland are key sources of nutrients in the RRW stream system. The extensive tile drain systems that have been established in the North Raccoon region are important conduits of nitrate to the RRW stream system.

[Figure 1]

The RRW is an intensively farmed region dominated by corn and soybean production. Cropland comprises about 79% of the watershed (Teshager et al., 2015) followed by pasture/grass (10%), developed areas (6%), mixed forest (4.4), and water bodies (0.5%). The watershed has a humid climate with both cold

and hot extremes, similar to most of the Midwest region. The average temperature in summer is about 22.7 °C and in winter is about -4.6 °C. Large variations in annual precipitation are very common. The annual precipitation varied from 606 mm in 1984 to 1372 mm in 1993, and the average annual precipitation was 829 mm, for the 30-year period of 1981 to 2010. About 75% of the precipitation falls in the months of April through September and peak monthly precipitation typically occurs within that period. Teshager et al.

(2015) estimated, based on data from Iowa Department of Natural Resources (IDNR), that about 57% of the watershed (~72% of the agricultural land) has tile drainage and 20% of the watershed receives manure application.

## 3  Simulations

### 3.1  Model Description and Setup

SWAT2012/Release 622 was the version of the model used for this study. SWAT is dynamic model that is typically executed on a daily time step although sub-daily options are also provided. The model is comprised of climate, soil, hydrology, management, nutrient cycling and transport, pesticide fate and



transport and several other components. Release 622 also features enhanced algorithms that account for more accurate representation of important switchgrass and miscanthus growth phenomena related to belowground biomass, plant respiration and nutrient uptake, which were developed by Trybula et al. (2015) and ported to standard SWAT versions starting with SWAT2012/release 615. A watershed is typically

delineated into subbasins in SWAT, based on topography, and each subbasin is then divided into multiple hydrological response units (HRUs) which consist of homogeneous soil, land use, topographic and management characteristics (Neitsch et al., 2011; Arnold et al., 2012). At present, HRUs are not spatially identified in applications of standard versions of SWAT although incorporation of expanded spatial detail is being developed (Duku et al., 2015; Arnold et al., 2010). Water and pollutants discharged at the HRU

level are input at the respective subbasin outlet and routed through the stream system to the watershed outlet. Neitsch et al. (2011) and Arnold et al. (2012) provide additional details about specific SWAT components, functions and/or input data requirements.

Baseline model testing (Teshager et al., 2015) was performed using ten weather stations distributed fairly uniformly across the watershed, and streamflow and in-stream pollutant data measured at a gauge

located near Van Meter, which drains 95% of the RRW. The model was calibrated and validated for the RRW for the years 2002 to 2010 for flow, total suspended solids (TSS), nitrate ($NO_3$) and mineral phosphorus (MINP) at daily, monthly and annual time scales (Teshager et al., 2015). Land use/land cover (LULC) from the USDA Cropland Data Layer (CDL; USDA-NASS, 2012) for the years 2002 to 2010 was used to develop crop rotations for calibration/validation of the watershed. According to Teshager et al.

(2015), about 14% of the watershed was planted in continuous corn (CC), 30% was in three-year rotations with one year of soybean and two years of corn (CCS/CSC/SCC), 31% was in two-year corn-soybean rotations (CS/SC), 6% was in three years rotations consisting of two years of soybean and one of corn (SSC/SCS/CSS), and 10% was pasture/grass (Fig. 1). The rest of the watershed included developed areas, forest or water bodies. The SWAT model was able to replicate flow, TSS, $NO_3$ and MINP satisfactorily at

daily, monthly and annual time scales.

### 3.2  Agricultural Scenarios

The most common approach in assessing climate change impacts is scenario construction (Öborn et al., 2011). The objective of a specific scenario, and the subject's complexity and time horizon shapes the method chosen for constructing scenarios (Dreborg, 2004). Due to the absence of a direct method for

predicting future farming choices, the agricultural scenarios developed in this study were developed based mainly on the need for more corn production for food, livestock feed and biofuel production, and the promising potential of switchgrass (SWG) for bio-energy production (Khanna et al., 2008; Schmer et al.,



2008; Secchi et al., 2008; Vadas et al., 2008). Accordingly, five agricultural scenarios were considered for the overall impact analysis (Table 1).

[Table 1]

The first scenario considered in this study assumed that future agricultural land use (crop type and rotation) matches historical agricultural land use patterns and is referred to as the baseline (BL) scenario. In addition to crop types and rotations, fertilizer/manure applications, tillage practices and tile drainage were

held constant through all three future simulation periods. Hence, the distributions of crop rotations described in the "Model set-up" section and Fig. 1 along with the management practices stated in Table 2 were used for the BL simulations.

[Table 2]

The second scenario reflects projections developed by the U.S. Department of Agriculture (USDA) that demand for corn will increase in the future based on an analysis of the world's agricultural sector in general and the U.S. agricultural sector in particular for the next decade (USDA, 2015). According to this

report, U.S. corn acreage is projected to remain high and production to rise gradually taking all uses of corn in to account. Thus this scenario is termed partial-corn (PC) and is simulated by converting selected HRUs into CC, as a function of baseline crop rotation, land use, and topographical conditions, to accommodate the projected increase in corn production. All baseline CCS/CSC/SCC rotations were converted to CC, due to the fact that those land parcels were already managed with relatively intense corn production. Next,

pasture HRUs with an average slope less than or equal to the current maximum cropland average slope were converted to CC; the slope constraint prevented conversion of extremely high sloped pasture land. About 52% of the watershed was planted in CC for this scenario, CS/SC and SSC/SCS/CSS rotations percentages remained the same, and about 2% of the watershed was still under pasture (Fig. 2a). Fertilizer applications to corn for CC cropping systems was 202 kg N/ha and 65 kg P/ha (as recommended by Duffy,

2013), in combination with conventional tillage, for the HRUs that were changed from other rotations or land uses to the CC rotation. The presence of tile drainage was held constant relative to the   baseline scenario.

[Figure 2]



The third scenario reflects adoption of switchgrass on selected RRW HRUs and is called the partial switchgrass (PS) scenario. The HRUs selected for this scenario were chosen based on baseline land use and topographical conditions. First, all pasture HRUs in the baseline scenario were converted to switchgrass. Moreover, cropland HRUs with average slope of greater than or equal to the average slope of pasture in the

baseline were changed to switchgrass, to maximize environmental benefits of converted cropland. Accordingly, about 41% of the watershed was converted to switchgrass in this scenario, resulting in decreases of 29%, 34%, 42% and 69% in CC, CCS/CSC/SCC, CS/SC and SSC/SCS/CSS relative to the BL scenario. As a result, about 10%, 19%, 18% and 2% of the remaining cropland was partitioned between CC, CCS/CSC/SCC, CS/SC and SSC/SCS/CSS, respectively (Fig. 2b). A nitrogen fertilizer application of

90 kg/ha was simulated for all converted cropland planted to switchgrass based on recommendations by Duffy (2008), Schumer et al. (2008), and McLaughlin and Kszos (2005). Tillage practices are not part of a perennial switchgrass cropping system and thus no till was the simulated tillage level by default. The PS scenario criteria underscore that the most productive corn-dominated cropland is located in very low slope areas.

The final two scenarios feature extreme conversions of all cropland and pasture land, representing 90% of the RRW, to either CC (all-corn scenario or AC) or switchgrass (all-switchgrass or AS). The same respective fertilizer and tillage assumptions described for the PC and PS scenarios were also used for these two scenarios.

The last two scenarios bracket hypothetical extreme future land use changes in the watershed and

represent the extent of the possible trade offs in food and fuel production, water quality and water quantity. The two partial scenarios are more realistic, and illustrate potential land use changes at a very fine resolution associated with climate change, global market forces, and energy and conservation policies. For example, the PS scenario could be associated with very aggressive climate mitigation and conservation policies, and the effective deployment of cellulosic ethanol and the corresponding phasing of corn ethanol.

**3.3 Climate Projections**

The climate projections were developed by downscaling output from multiple coupled atmospheric-ocean general circulation models (AOGCMs) to the locations of watershed weather stations. AOGCMs represent the primarily tools available to assess the large scale climatic response to changes in forcing, such as the expected changes in 21st century greenhouse gas concentrations. In this study, eight AOGCMs

(Table 3), which were all included in Phase 5 of the Coupled Model Intercomparison Project (CMIP5; Taylor et al., 2012), were utilized in developing climate change projections for the RRW land use change scenario simulations. Using AOGCM ensembles incorporates information from different models, often



increasing the value of the climate information obtained (Knutti et al. 2010; Martre et al. 2015; Pierce et al., 2009; Weigel et al. 2010) and thus an improved overall climate change impact analysis.

Each of these eight climate models were forced with three representative concentration pathways (RCPs) representing low (RCP2.6), medium (RCP4.5) and high (RCP 8.5) levels of radiative forcing from

GHGs (Moss et al., 2010; van Vuuren et al., 2011). The RCP2.6 pathway depicts future conditions which represent "medium development" of global population, income, and energy use and land use, resulting in a peak atmospheric $CO_2$ concentration prior to 2100 (van Vuuren et al., 2011a; 2011b). A cost-minimizing approach is used in the RCP4.5 pathway, which assumes that simultaneous efforts occur worldwide to mitigate emissions, including taking into account the cost of reducing emissions per the 100-year warming

potential of a respective GHG, resulting in stabilization of atmospheric $CO_2$ concentrations in 2100 (Thomson et al., 2011; van Vuuren et al., 2011a). High energy demand and GHG emissions characterize the RCP8.5 pathway, which occur due to assumed high population and slow income growth with modest rates of technological change and energy improvement, without implementation of climate change adaptation policies (Riahi et al., 2011; van Vuuren et al., 2011a).

The PC and AC agricultural scenarios reflect land use patterns, management systems and energy use levels that could potentially contribute to higher GHG emissions (Davis et al., 2012), that would be consistent with the RCP8.5 pathway. Planting switchgrass, on the other hand, has a potential to sequester carbon (Keshwani and Cheng, 2009; Davis et al., 2012) and help reduce $CO_2$ emission in the long term. Thus the AC scenario could be viewed as being consistent with the RCP8.5 pathway and the AS scenario

could be considered as a system consistent with the RCP2.6 pathway, due to expected lower GHG emissions that would occur during the next century to the expanded switchgrass production. These hypothetical relationships between the future agricultural scenarios and the RCP pathways are investigated to some extent per the interactions of different agricultural scenarios and climate projections in the results section.

[Table 3]

Contemporary AOGCMs are archived with a resolution of approximately 2°, although there is substantial variability in model resolution among participating modeling groups. To conduct impact analysis using models like SWAT, higher resolution information is required. Thus, downscaling to a finer

resolution is crucial to incorporate local climate variability for detailed watershed assessments. Here, a statistical downscaling approach involving regression-based models and stochastic weather simulation, as described by Schoof et al. (2007) and Schoof (2015) was used to derive station-based projections consistent with the projections of the parent AOGCMs under each emissions pathway. These downscaled climate data were then post-processed to produce a comprehensive daily weather dataset (precipitation, minimum and



maximum temperature, relative humidity, solar radiation, and wind speed) for the years 2011 to 2100 to be used in the SWAT model scenario simulations.

In addition to the three emission scenarios (RCPs), the weather data were divided in to three temporal blocks of 20 years to represent early (2016-2035), mid (2046-2065) and late (2076-2095) century

climate conditions. As a result, a total of 72 (8 climate models × 3 emission scenarios × 3 temporal scenarios) climate scenarios were created. Moreover, simulating climate change scenarios in SWAT requires the $CO_2$ concentration for the simulation time periods. Accordingly a single average value of $CO_2$ concentration was used in simulating each 20-year temporal block, similar to the approach used by Ficklin et al. (2009), for a given RCP scenario (Table 4). These scenarios were used to run simulations through the

calibrated SWAT model for each agricultural scenario discussed in the "Model set-up" section at the annual time scale.

[Table 4]

**3.4    Method of Analysis**

Reporting SWAT output values for each year was not feasible due to the fact that 360 total land use change and climate change combinations (72 climate × 5 agricultural scenarios) were simulated in the study. Therefore, annual average and standard deviation values for each temporal block (early, mid and late century), RCP pathway (2.6, 4.5 and 8.5) and agricultural land use change scenario were reported for each

output indicator of interest: stream flow (Q), total suspended solids (TSS), total nitrogen (TN), and total phosphorous (TP). This approach allowed us to capture both the trends across temporal blocks and agricultural scenarios, and variations within temporal blocks and across climate models. Moreover, the predicted average corn and switchgrass yields were also determined for each temporal block (consisting of eight climate models) for the AC and AS agricultural scenarios, respectively.

**4    Results and Discussions**

**4.1    Weather**

Table 5 shows a comparison between historical observed and future projected average annual precipitation and annual average temperature values along with standard deviations across the years and among AOGCMS. The results show that, on average, annual precipitation and temperature values increase

from early to late century (and from RCP2.6 to RCP8.5). Compared to the average historical observations between years 1991 and 2010, the annual average temperature values for the RCP2.6, RCP4.5 and RCP8.5 pathways within the early, mid and late century time periods all increased by 1.5 to 4.2 °C (Fig. 3),



depending on the RCP and time period. In contrast, there were decreases in average annual precipitation values for all of the scenarios except the late-century RCP4.5 and RCP 8.5 scenarios (Fig. 3).

[Figure 3]

Similar results have been reported in previous studies. Chien et al. (2013) reported that, compared to 1990-1999, the average temperature increased by up to ~3ºC (~5ºC) for 2051-2060 (2086-2095) and the percentage change in annual precipitations were about -28% to +8% (-33% to +16%) for 2051-2060 (2086-2095), using data from nine GCMs for four watersheds, which cover portions of Illinois, Indiana and Wisconsin. Similarly, Ficklin et al. (2012) analyzed downscaled temperature and precipitation projections from 16 GCMs (two emission scenarios, low (B1)  and high (A2)) for Mono Lake basin, California, and found that the 2070-2099 annual average temperature increased by 2.5ºC and 4.1ºC for B1 and A2 scenarios, respectively, compared to 1961-1990. However, they also reported that there was a slight but statistically insignificant decrease in annual precipitation on average. These previous studies confirm the results found here, that there is a consistent trend of increases in temperature across climate models and geographical locations, while precipitation could increase or decrease depending on the choice of AOGCMs, projection pathway and geographical location of the analysis.

[Table 5]

The interannual variation (standard deviation) was much higher (≤ factor of 4) for the historical observed temperature and precipitation versus the corresponding future projections (Table 5). The variations among climate models increased for both temperature and precipitation from early to late century. Moreover, the standard deviations among AOGCMs were higher than (≥ factor of 2) the interannual variations for annual average temperature values. Chapman & Walsh (2006) found similar differences between models and interannual variabilities (standard deviation) of temperature using 14 AOGCMs. For average annual precipitation values, the standard deviations among AOGCMs were slightly higher than interannual variations. These results were mainly due to consideration of an ensemble of AOGCMs that has an effect of reducing interannual variations compared to interannual variations from individual AOGCMs (Knutti et al., 2010). Therefore, one should take into account these effects in using ensembles of AOGCM results for impact analysis. Moreover, variations among AOGCMs may indicate that the choice of models within an ensemble for climate change impact analysis may result in different conclusions.



### 4.2 Stream Flow (Q)

The historical (1991 to 2010) annual average Q at the watershed outlet was about 212 mm (63 m$^3$/s). There were both predicted decreases (1% to 24%) and increases (3% to 75%) in Q for the BL, AC, and PC agricultural scenarios in response to the different climate projections (Fig. 4a-c), relative to the historical

average Q. For the PS and AS scenarios, however, there were decreases (15% to 83%) in Q for all but one of the climate projections (Fig. 4a-c). Despite decreases in precipitation and increases in temperature, an increase in Q in some of the scenarios indicates the possible occurrence of larger and more frequent high intensity precipitation events than the historical observed values in the projected climate data (Schoof, 2015; Kharin and Zwiers, 2000). Moreover, a reduction in ET due to increased $CO_2$ levels, especially in the

mid and late century periods, also contributed to simulated increases in stream flow in mid- and late-century scenarios similar to results reported by Jha et al. (2006) and Wu et al. (2012).

The PS and AS scenarios resulted in lower estimated Q compared to the other scenarios and the historical baseline, for a given climate scenario, while very small difference were observed between the BL, AC and PC scenarios (Fig. 4a-c). The AS agricultural scenario exhibited the highest decrease in stream

flow (or water yield) as expected. Similar results were indicated by previous studies (e.g., Kim et al., 2013; Parajuli and Duffy, 2013; Schilling et al., 2008; Wu et al., 2013). This reveals that large-scale conversion to switchgrass could result in reduced water availability due to increased ET and conversely reduced Q, which could render it less desirable as a climate change adaptation strategy in the watershed for future climate conditions that manifest lower precipitation levels. Also, as noted previously, the AC and AS

scenarios reflect agricultural production schemes that are consistent with the high GHG emission RCP8.5 pathway or the low emission RCP2.6 pathway, respectively. A comparison on this basis reveals that the AS scenario resulted in a much higher reduction in Q compared to the AC scenario, relative to the previous comparison (Fig. 4a-c), which further underscores that widespread adoption of just switchgrass in current intensively cropped Corn Belt watersheds may not be a viable strategy in mitigating climate change

impacts on water availability.

Comparisons were also made between climatic projections for a given agricultural scenario. The results show a decrease in Q relative to historical observed values for early-century under all RCPs (Fig. 4a). At mid-century, decreases in Q were predicted for the majority of agricultural scenario-climate

projection combinations, except for the BL, AC, and PC scenarios in response to the projected RCP8.5 pathway. However, there was a consistent increase in Q, during the late century time period, across agricultural scenarios in response to the RCP8.5 projection and for the BL, AC, and PC scenarios when impacted by the RCP4.5 projection (Fig. 4a-c). These increases in Q from early-to-late century could be attributed to the precipitation increase in the same manner as discussed in Section 4.1. Except for the early



century time period, Q increased from RCP2.6 to RCP8.5 for all agricultural scenarios. The maximum increases (≤ 75% of historical Q) were simulated under the late-century RCP8.5 for all agricultural scenarios.

Previously, Jha and Gassman (2013) used an ensemble of GCMs projections, developed within the framework of CMIP Phase 3 (CMIP3; PCMDI, 2016), to simulate the impacts of projected future climate change on the RRW with SWAT. They concluded that there was an overall average decrease in total Q of 17% in the mid-century period, compared to Q for the years 1961 to 2000. Similar BL scenario results were obtained in this study for the RCP2.6 projections (14.7%, 11.0% and 14.7 % for the early, mid and late 21st century, respectively) and RCP4.5 and RCP8.5 early-century projections (23.7% and 12.4%, respectively). The mid-century RCP4.5 scenarios showed a slight decrease (2.6%) in Q while increases in Q were simulated for the RCP8.5 scenario in both the mid and late century (12.3% and 72.6%, respectively), and for the late-century RCP4.5 scenario (9%).

The standard deviations of annual Q between AOGCMs and future time periods (Fig. 4 and Tables A1 & A2) followed trends similar to the temperature and precipitation results discussed in Section 4.1. The standard deviation across time periods for the historical period was greater than for any of the future temporal periods for all of the agricultural scenarios. Similarly, the standard deviation between AOGCMs is greater than that across future time periods. These trends are also similar for all TSS, TN and TP values (Fig. 4 and Tables A1 & A2).

## 4.3 Total Suspended Sediment (TSS)

Simulated TSS impacts for the different agricultural scenario-climate projection combinations were compared to each other and versus the simulated historical TSS values. The historical (1991 to 2010) annual average TSS concentration at the watershed outlet was about 113 mg/L (2.25 x $10^5$ metric ton). Compared to the historical TSS concentration, there were increases in TSS for the AC and PC scenarios across all climate projections, decreases for the PS scenario for most of the climate projections and decreases for AS in all three climate projections (Fig. 4d-f). The increases in TSS were highest for the AC (≤ 67%) scenario, followed by the PC (≤ 65%) and BL (≤ 63%) scenarios. Peak TSS decreases were 74% and 27% for the AS and PS scenarios, respectively.

For a given climate scenario, there were 1.6% to 7.1% increases in TSS for PC and 2.3% to 11.1% increases for AC compared to the BL scenario. This indicates how intensively the RRW is utilized for agricultural production already. It was only when switchgrass was introduced (AS and PS scenarios) that significant decreases in TSS were observed (18% to 27% for PS and 56% to 74% for AS) relative to the BL scenario. Hence, switchgrass seems to be a good adaptation strategy with respect to addressing TSS reductions. This result is magnified when results are assessed based on agricultural scenarios simulated





with the appropriate climate scenarios, as discussed in "Stream Flow (Q)" section. Generally, the predicted TSS values followed the Q trends for all of the climate projection and agricultural scenario categories (Fig. 4d-f). For a given agricultural scenario, TSS increased continuously from the early-to-late 21st century. Considerable reduction in TSS was simulated in the AS agricultural scenario under all climate scenarios

compared to historical levels. However, the AS scenario must be viewed as extreme and impractical, due to the importance of corn as a crop in the RRW and Corn Belt region in general. However, the PS agricultural scenario, which is a more plausible scenario, may require additional best management practices to significantly reduce TSS yield and transport from the watershed.

[Figure 4]

## 4.4 Total Phosphorous (TP)

The annual average historical (1991 to 2010) simulated TP at the watershed outlet was roughly $4.52 \times 10^3$ metric tons (or 7.6 mg/L). Comparisons were made between the different scenario results, and

between historical and scenario results. Due mainly to the absence of phosphorus fertilizer application and reduction in surface runoff when planting switchgrass, there were significant reductions in TP in the PS and AS agricultural scenarios compared to historical simulated values (up to 66% and 99%, respectively) and BL scenario (up to 49% and 99%, respectively) (Fig. 4d-f). The differences in tillage practices between agricultural scenarios also contributed to the difference in TP output among scenarios, due to the shifts in

tillage practices used in the BL scenario versus just conventional tillage for CC in the PC and AC scenarios, and elimination of tillage for the PS and AS scenarios.  Conventional tillage practices result in higher sediment and phosphorus yields but conservation and no-till tillage practices can result in lower yields under some conditions. Various researchers (e.g., Parajuli et al., 2013; Tomer et al. 2008; Andraski et al., 2003; Bundy et al., 2001) have demonstrated similar effects of tillage practices on sediment and/or

phosphorous outputs from agricultural fields.

For a given climate scenario, the PS and AS scenarios exhibited similar reductions in TP output ≤ 49% and 99%, respectively) compared to the BL scenario. Both of the CC-based scenarios (PC and AC) resulted in large increases in TP, compared to both the BL scenario for all climate projections (≤ 36% and 41% for PC and AC, respectively) and historical simulated values (≤ 62% and 67% for PC and AC,

respectively).

## 4.5 Total Nitrogen (TN)

The annual average historical (1991 to 2010) simulated TN load value at the watershed outlet was about $2.14 \times 10^4$ metric ton (or 36 mg/L). Comparisons were made between simulated historical and scenario annual average TN load values at the watershed outlet, and also among scenarios. These comparisons





reveal two important insights: (1) the AC scenario resulted in lower TN loads relative to the BL scenario, which was not originally expected, and (2) the PC scenario resulted in highest TN loads of all of the agricultural scenarios (Fig. 4a-c). This implies that, with respect to TN output, the current agricultural management conditions (BL scenario) in the RRW are already extremely intensive, and are comparable to

planting continuous corn everywhere with conventional tillage and 202 N kg/ha of fertilizer (AC scenario). Even though the fertilizer application rates were less than 202 N kg/ha in the BL scenario, manure was applied in addition to the fertilizer (Teshager et al., 2015), resulting in a slightly higher TN load for the BL scenario. However, as previously described the PC scenario reflects a combination of BL scenario cropping system and management practices in combination with increased conversion of some land parcels to CC

resulting in slightly higher TN loads as compared to both the BL and AC scenarios. Also, the introduction of switchgrass in the RRW AS and PS scenarios has the potential to reduce the total nitrogen outflow from the watershed significantly  relative to historical levels ($\leq$ 84% for AS and $\leq$ 35% for PS) as shown in Fig. 4a-c. For a given climate projection, annual average TN loads were reduced up to 81% for AS and up to 18% for the PS scenarios in comparison to the BL scenario. This was due to the both the elimination of

tillage in switchgrass cropping systems and the capability of switchgrass to scavenge nitrate from the soil-water matrix. Planting switchgrass in select areas of a watershed, similar to the PS scenario approach, and implementing effective best management practices could further reduce nitrogen losses to Corn Belt stream systems.  The effects of expanded adoption of switchgrass depicted in the PS and AS scenarios on reductions in TN loads are further magnified when examining the results within the context of the RCP4.5

and RCP2.6 pathways, which were previously identified as the two respective pathways that the PS and AS scenarios were most correlated with, especially for the late century time period. Similar to the Q and TSS results, the TN loads increased from the early part of the century to the late part of the century, especially for the RCP4.5 and RCP8.5 pathways (Fig. 4a-c).

### 4.6   Crop Yields

Crop yield analyses were done to point out the potential impacts of climate change on corn and switchgrass yields, assuming that the current production technologies for both crops remain the same, based on crop yield estimates obtained from the AC and AS scenarios. The 20-year (1991 to 2010) historical simulated average yields across the entire RRW was 10 t/ha for corn and 15.5 t/ha for switchgrass. The AC scenario corn yields and AS scenario switchgrass yields were predicted to be decline

across future climate conditions, as compared to the historical simulated yields, especially during the mid and late centuries for the higher RCP4.5 and RCP8.5 GHG emission pathways (Fig. 5).

[Figure 5]





The reduction in corn yields ranged from 7% during the early century time period to 25% in the late-century time period (Fig. 5). However, no reductions were predicted for switchgrass yields initially in the early century but estimated declines in switchgrass yields of ≤ 19% occurred in the latter part of the century. In the early century, the effects of the emission pathways on the crop yields were insignificant;

however, the emission pathway effects became more pronounced in the mid- and late-century simulations (Fig. 5). There were essentially no differences in corn or switchgrass yields between the early, mid and late century time period simulations for the low emission RCP2.6 pathway. The highest yield reductions, 25% for corn and 19% for switchgrass, were simulated in response to the high emission RCP8.5 pathway at the end of the century. Lower percentage crop yield reductions were found in this study compared to similar

previous research results (e.g., Miao et al., 2015; Ummenhofer et al., 2013; Cai et al., 2009; Schlenker and Roberts, 2008). One possible reason that lower reductions in crop yields were predicted within this study could be the inclusion of $CO_2$ concentrations during the simulations and the capability of SWAT to account for positive effects of $CO_2$ concentration on crop yield. In addition, higher precipitation amounts that characterize the RCP8.5 pathway late century time period could have partly offset the effects of increased

temperatures on yield. However, the predicted corn and switchgrass yields for the RCP8.5 pathway late century time period were lower than other time periods, even though the average annual precipitation was higher than the historical or any other future projected precipitation. This result is consistent with the results presented in Section 4.2 because the increase in annual precipitation was due mainly to more high intensity daily precipitation events (Schoof, 2015), which will not necessarily be beneficial for crop growth.

**5  Conclusions and Recommendations**

The SWAT simulation results representing five agricultural scenarios, eight AOGCMs, three representative concentration pathways (RCP2.6, RCP4.5 and RCP8.5) and three twenty-year temporal blocks (early, mid, and late 21st centuries) were systematically aggregated to analyze the combined impacts of agricultural scenario and climate change on water, total suspended solids, total nitrogen and total

phosphorous yields at the Raccoon River watershed outlet. Moreover, the effects of climate change on corn and switchgrass yields were assessed by analyzing the results of the AC and AS scenarios.

In general, the results indicated the need for developing alternative biofuel cropping systems to counteract future problems that could develop from relying on intensification of corn production in Corn Belt region watersheds to mitigate potential future water quality problems. The results of this study were

consistent with the findings of Wilson and Weng (2011), that future climate change would exert a larger impact on the concentration of pollutants than the potential impact of land use (Fig. 4a-f). The results also showed that significant reduction in water pollution could be accomplished by expanded planting of switchgrass in the RRW as depicted by the PS and AS scenarios. Even though it provides the best results in



alleviating water quality problems in the future, the promising future water quality benefits suggested by the AS scenario results are unrealistic due to the need for production of corn or other crops. Moreover, there were scenarios where results indicated reductions in water quality in PS relative to the BL historical simulation. This shows that planting switchgrass alone may not be sufficient to improve water quality for

5    watersheds like RRW.

Therefore, future work will focus on using the different climate scenarios to assess how implementing best management practices, such as cover crops, less intensive tillage practices, fertilizer application timing and amount, filter strips, etc., in addition to planting switchgrass partially on selected lands, performs in reducing water pollution from agricultural lands. Moreover, monthly analysis, similar to that of Jha et al.

10   (2006) and Jha and Gassman (2014), could reveal additional results more relevant for water resources in watersheds like RRW where the river is utilized for municipal and industrial water supply purposes.

**Appendix A: Supplemental Tables**

[Table A1]

[Table A2]

**Acknowledgments**

This material is based upon work supported by the National Science Foundation under Grant No.

20   1009925. Any opinions, findings, and conclusions or recommendations expressed in this material are those of the authors and do not necessarily reflect the views of the National Science Foundation.



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



Table 1: Percentage of crop rotations and LULC in each agricultural scenario considered (BL=Baseline, PC=Partial Corn, AC=All Corn, PS=Partial Switchgrass, AS=All Switchgrass)

| Agricultural Scenario | CC | CCS/ CSC/ SCC | CS | SSC/ SCS/ CSS | SWG | PAST | FRST | WATR | URHD |
|---|---|---|---|---|---|---|---|---|---|
| BL | 13.8 | 29.0 | 30.6 | 5.8 | 0.0 | 10.0 | 4.4 | 0.5 | 5.9 |
| PC | **51.3** | 0.0 | 30.6 | 5.8 | 0.0 | 1.5 | 4.4 | 0.5 | 5.9 |
| AC | **89.2** | 0.0 | 0.0 | 0.0 | 0.0 | 0.0 | 4.4 | 0.5 | 5.9 |
| PS | 9.8 | 18.6 | 18.0 | 1.7 | **41.1** | 0.0 | 4.4 | 0.5 | 5.9 |
| AS | 0.0 | 0.0 | 0.0 | 0.0 | **89.2** | 0.0 | 4.4 | 0.5 | 5.9 |

Table 2 Fertilizer/manure application rates and presence of tiles and tillage practices (SOYB=soybeans, NT=No-till, Cs=conservation tillage, Cv=conventional tillage)

| Crop Type | Rotation | Fertilizer | | Manure | Tile | Tillage |
|---|---|---|---|---|---|---|
| | | kg N/ha | kg P/ha | (kg N/ha) | | |
| CORN | CORN after CORN | 165 | 65 | 179 | Yes | NT, Cs, Cv |
| | CORN after SOYB | 150 | 70 | | | |
| SOYB | SOYB after CORN | 15 | 55 | 0 | | |
| | SOYB after SOYB | 0 | 0 | | | |

*Source: Teshager et al. (2015)*



Table 3: AOGCMs considered in this study

| Model Name | Modeling Center (or group) | Reference |
|---|---|---|
| BCC-CSM1 | Beijing Climate Center, China Meteorological Administration | Wu et al., 2010 |
| BNU-ESM | College of Global Change and Earth System Science, Beijing Normal University | Ji et al., 2014 |
| CanESM2 | Canadian Centre for Climate Modelling and Analysis | Chylek et al., 2011 |
| CNRM-CM5 | Centre National de Recherches Meteorologiques/ Centre Europeen de Recherche et Formation Avancees en Calcul Scientifique | Voldoire et al., 2013 |
| IPSL-CM5A | Institut Pierre–Simon Laplace | Dufresne et al., 2013 |
| MPI-ESM | Max Planck Institute for Meteorology | Jungclaus et al., 2010 |
| MRI-CGCM3 | Meteorological Research Institute | Yukimoto et al., 2012 |
| NOR-ESM | Norwegian Climate Centre | Kirkevåg et al., 2008 |

Table 4: Carbon dioxide concentration (ppm) values used in SWAT simulations

| Scenario | Early-century | Mid-century | Late-century |
|---|---|---|---|
| RCP2.6 | 418 | 441 | 429 |
| RCP4.5 | 424 | 495 | 532 |
| RCP8.5 | 436 | 578 | 804 |





Table 5: Mean and standard deviations of average annual temperature and precipitation values for historical observed and ensembles eight climate models used in this study ($T_{avg}$=annual average temperature, PCP=average annual precipitation $T_{avg,std}$=standard deviation of annual average temperature, $PCP_{std}$=standard deviation of average annual precipitation)

| Century | Scenario | $T_{avg}$ (ºC) | PCP (mm) | Among 8 Models | | Across 20 Years | |
|---|---|---|---|---|---|---|---|
| | | | | $T_{avg,std}$ | $PCP_{std}$ | $T_{avg,std}$ | $PCP_{std}$ |
| **Historical** (1991-2010) | Observed | 9.2 | 831.1 | NA | NA | 0.77 | 175.7 |
| **Early** (2016-2035) | RCP26 | 10.7 | 806.3 | 0.42 | 49.2 | 0.25 | 49.1 |
| | RCP45 | 10.8 | 791.0 | 0.43 | 52.3 | 0.27 | 49.7 |
| | RCP85 | 10.8 | 808.4 | 0.43 | 52.0 | 0.26 | 49.2 |
| **Mid** (2046-2065) | RCP26 | 11.1 | 816.4 | 0.42 | 56.4 | 0.20 | 50.7 |
| | RCP45 | 11.5 | 820.3 | 0.54 | 59.9 | 0.27 | 46.5 |
| | RCP85 | 12.0 | 827.1 | 0.51 | 67.2 | 0.33 | 49.9 |
| **Late** (2076-2095) | RCP26 | 11.1 | 813.6 | 0.52 | 54.5 | 0.24 | 55.6 |
| | RCP45 | 11.9 | 831.1 | 0.55 | 62.2 | 0.24 | 52.7 |
| | RCP85 | 13.4 | 868.9 | 0.72 | 83.2 | 0.32 | 57.7 |



Table A1: Standard deviation of Q, TSS, TN and TP among 8 climate models

| | LULC | Early | | | Mid | | | Late | | |
|---|---|---|---|---|---|---|---|---|---|---|
| | | RCP2.6 | RCP4.5 | RCP8.5 | RCP2.6 | RCP4.5 | RCP8.5 | RCP2.6 | RCP4.5 | RCP8.5 |
| **Flow, Q (mm)** | **BL** | 37.70 | 36.46 | 35.79 | 42.06 | 55.28 | 53.16 | 40.79 | 59.33 | 81.05 |
| | **AC** | 38.10 | 38.90 | 36.60 | 43.13 | 55.04 | 52.94 | 41.71 | 55.14 | 81.54 |
| | **PC** | 37.91 | 36.60 | 36.14 | 42.28 | 55.23 | 53.23 | 41.22 | 55.77 | 81.99 |
| | **PS** | 34.18 | 30.50 | 31.34 | 38.00 | 50.60 | 49.41 | 35.91 | 53.38 | 84.09 |
| | **AS** | 27.05 | 19.30 | 21.43 | 28.57 | 45.45 | 42.59 | 26.74 | 48.35 | 87.64 |
| **TSS (mg/L)** | **BL** | 16.35 | 16.58 | 15.94 | 18.33 | 22.21 | 20.38 | 17.70 | 22.50 | 24.52 |
| | **AC** | 16.73 | 17.69 | 16.73 | 18.18 | 20.53 | 26.23 | 18.67 | 23.14 | 21.47 |
| | **PC** | 17.22 | 17.51 | 16.90 | 18.84 | 23.80 | 22.00 | 18.62 | 21.20 | 26.36 |
| | **PS** | 15.33 | 15.62 | 15.35 | 18.24 | 19.96 | 19.34 | 17.04 | 18.72 | 21.21 |
| | **AS** | 10.72 | 12.85 | 12.19 | 15.13 | 15.84 | 14.67 | 13.51 | 15.96 | 18.20 |
| **TN (1000 ton)** | **BL** | 3.76 | 3.93 | 3.53 | 4.48 | 4.28 | 4.16 | 3.56 | 5.06 | 6.05 |
| | **AC** | 3.79 | 4.01 | 3.49 | 4.54 | 4.62 | 5.02 | 3.84 | 5.56 | 9.69 |
| | **PC** | 3.89 | 4.11 | 3.58 | 4.74 | 4.59 | 4.71 | 3.77 | 5.42 | 7.88 |
| | **PS** | 3.60 | 3.47 | 3.10 | 3.76 | 3.61 | 3.69 | 3.21 | 4.48 | 5.56 |
| | **AS** | 2.97 | 2.99 | 2.82 | 2.83 | 2.82 | 3.50 | 3.22 | 4.13 | 5.14 |
| **TP (1000 ton)** | **BL** | 0.83 | 0.82 | 0.85 | 1.02 | 0.98 | 1.03 | 0.72 | 1.20 | 1.88 |
| | **AC** | 1.22 | 1.23 | 1.25 | 1.45 | 1.40 | 1.50 | 1.09 | 1.73 | 2.52 |
| | **PC** | 1.19 | 1.17 | 1.19 | 1.39 | 1.35 | 1.43 | 1.05 | 1.64 | 2.36 |
| | **PS** | 0.41 | 0.42 | 0.45 | 0.56 | 0.52 | 0.53 | 0.44 | 0.62 | 1.14 |
| | **AS** | 0.016 | 0.011 | 0.015 | 0.015 | 0.023 | 0.016 | 0.013 | 0.022 | 0.025 |




Table A2: Standard deviation of Q, TSS, TN and TP across years (in each 20 years block)

|  | LULC | Early | | | Mid | | | Late | | | Historical |
|---|---|---|---|---|---|---|---|---|---|---|---|
|  |  | RCP2.6 | RCP4.5 | RCP8.5 | RCP2.6 | RCP4.5 | RCP8.5 | RCP2.6 | RCP4.5 | RCP8.5 |  |
| **Flow,Q (mm)** | **BL** | 33.97 | 34.58 | 30.88 | 35.89 | 33.24 | 37.45 | 37.54 | 37.87 | 47.76 | 107.50 |
|  | **AC** | 35.30 | 35.54 | 31.69 | 37.24 | 33.83 | 38.26 | 39.02 | 37.20 | 48.22 |  |
|  | **PC** | 34.82 | 34.83 | 31.13 | 36.39 | 33.36 | 37.84 | 38.41 | 37.16 | 48.03 |  |
|  | **PS** | 29.95 | 28.71 | 25.98 | 30.78 | 28.70 | 32.76 | 33.31 | 34.32 | 47.20 |  |
|  | **AS** | 18.36 | 16.40 | 17.02 | 16.33 | 19.50 | 24.15 | 20.48 | 28.85 | 45.75 |  |
| **TSS (mg/L)** | **BL** | 15.05 | 15.68 | 13.70 | 16.05 | 13.78 | 15.42 | 16.04 | 14.67 | 15.41 | 34.13 |
|  | **AC** | 15.72 | 16.42 | 14.47 | 16.49 | 14.32 | 15.97 | 16.72 | 14.12 | 15.55 |  |
|  | **PC** | 15.97 | 16.67 | 14.51 | 16.64 | 14.43 | 16.13 | 17.05 | 14.36 | 15.64 |  |
|  | **PS** | 14.24 | 14.45 | 12.52 | 15.47 | 12.72 | 14.38 | 15.83 | 13.30 | 13.89 |  |
|  | **AS** | 9.99 | 12.15 | 10.00 | 11.18 | 11.54 | 11.74 | 12.69 | 13.66 | 12.87 |  |
| **TN (1000 ton)** | **BL** | 3.84 | 3.83 | 3.36 | 4.03 | 3.89 | 3.84 | 3.89 | 4.15 | 4.15 | 14.47 |
|  | **AC** | 3.79 | 3.93 | 3.43 | 4.13 | 3.92 | 4.44 | 3.92 | 4.25 | 5.76 |  |
|  | **PC** | 3.88 | 4.01 | 3.46 | 4.27 | 3.89 | 4.20 | 3.94 | 4.26 | 4.95 |  |
|  | **PS** | 4.07 | 3.41 | 2.92 | 3.41 | 3.28 | 3.51 | 3.63 | 3.94 | 3.66 |  |
|  | **AS** | 6.21 | 3.88 | 3.82 | 4.01 | 3.55 | 3.62 | 4.66 | 5.93 | 4.13 |  |
| **TP (1000 ton)** | **BL** | 0.83 | 0.90 | 0.86 | 0.95 | 0.84 | 1.04 | 0.74 | 1.00 | 1.49 | 3.05 |
|  | **AC** | 1.20 | 1.35 | 1.34 | 1.35 | 1.22 | 1.51 | 1.13 | 1.53 | 1.92 |  |
|  | **PC** | 1.15 | 1.29 | 1.27 | 1.29 | 1.16 | 1.42 | 1.07 | 1.45 | 1.82 |  |
|  | **PS** | 0.44 | 0.43 | 0.42 | 0.51 | 0.43 | 0.53 | 0.45 | 0.54 | 0.80 |  |
|  | **AS** | 0.012 | 0.011 | 0.012 | 0.011 | 0.016 | 0.013 | 0.010 | 0.015 | 0.015 |  |





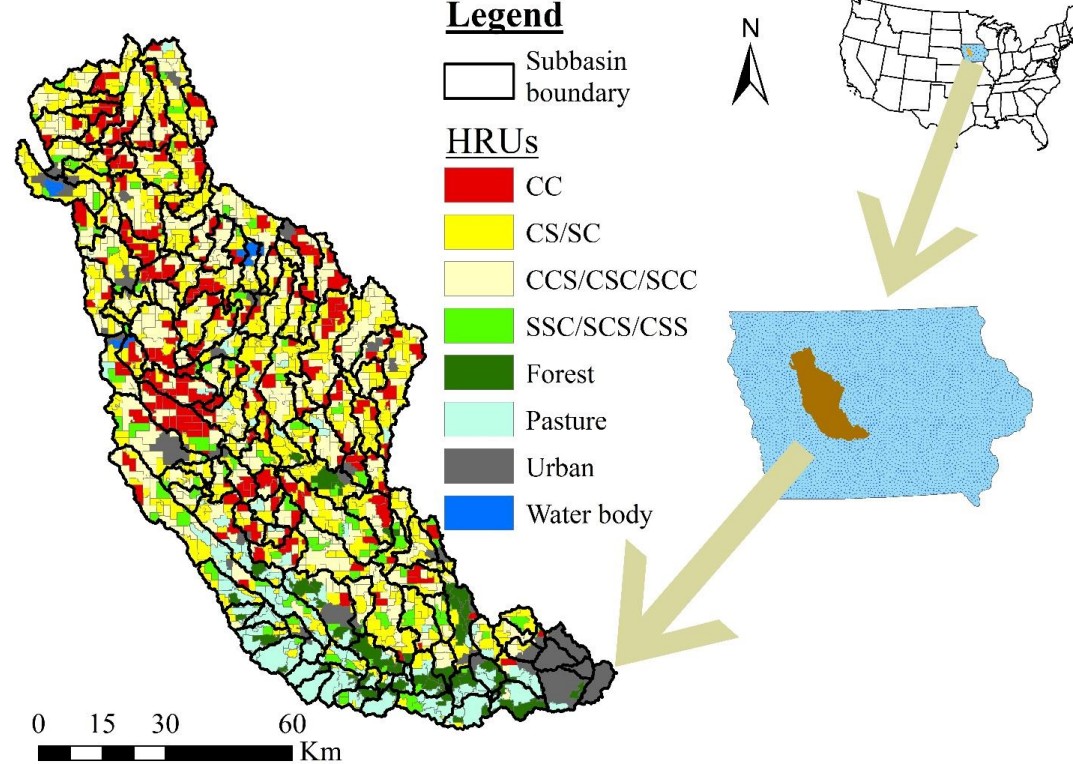

Figure 1: RRW with its historical (baseline) land use (CC=continuous corn rotation, CS/SC=corn-soybeans rotation, CCS/CSC/SCC=two years of corn and one year of soybeans in three years rotation, SSC/SCS/CSS=two years of soybeans and one year of corn in three years rotation)



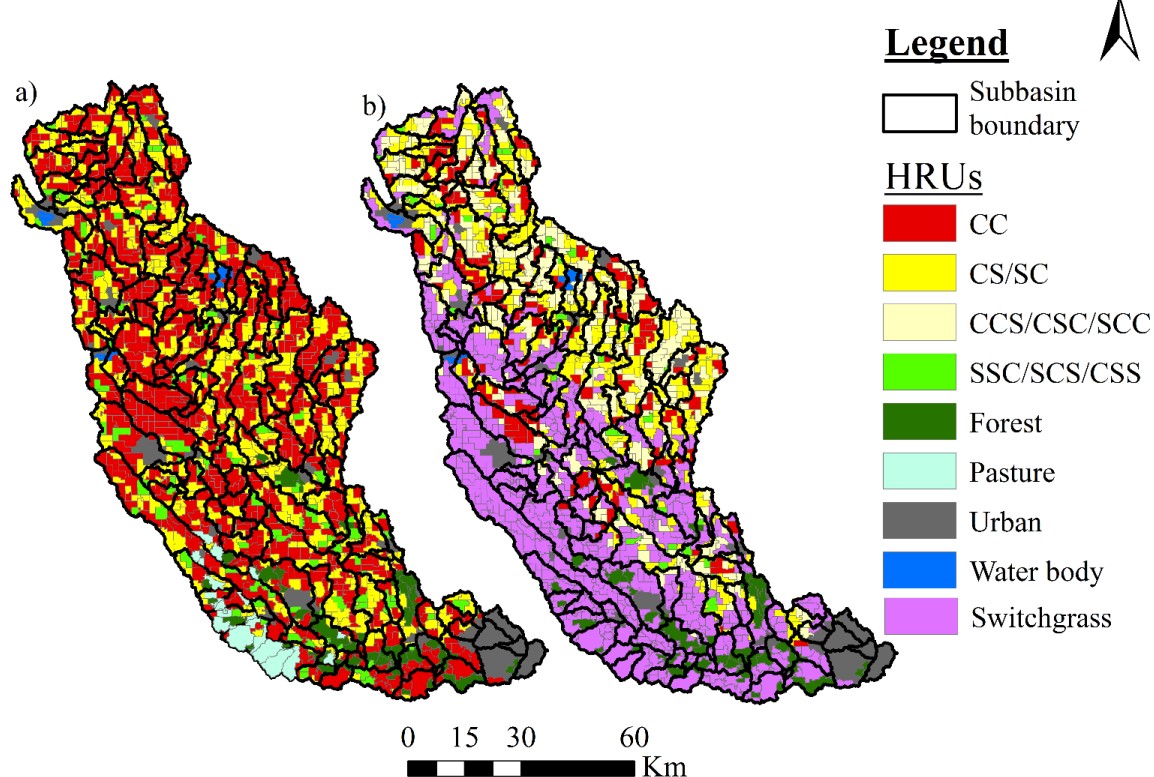

Figure 2: a) Partial-Corn and b) Partial-Switchgrass agricultural scenarios (CC=continuous corn rotation, CS/SC=corn-soybeans rotation, CCS/CSC/SCC=two years of corn and one year of soybeans in three years rotation, SSC/SCS/CSS=two years of soybeans and one year of corn in three years rotation)




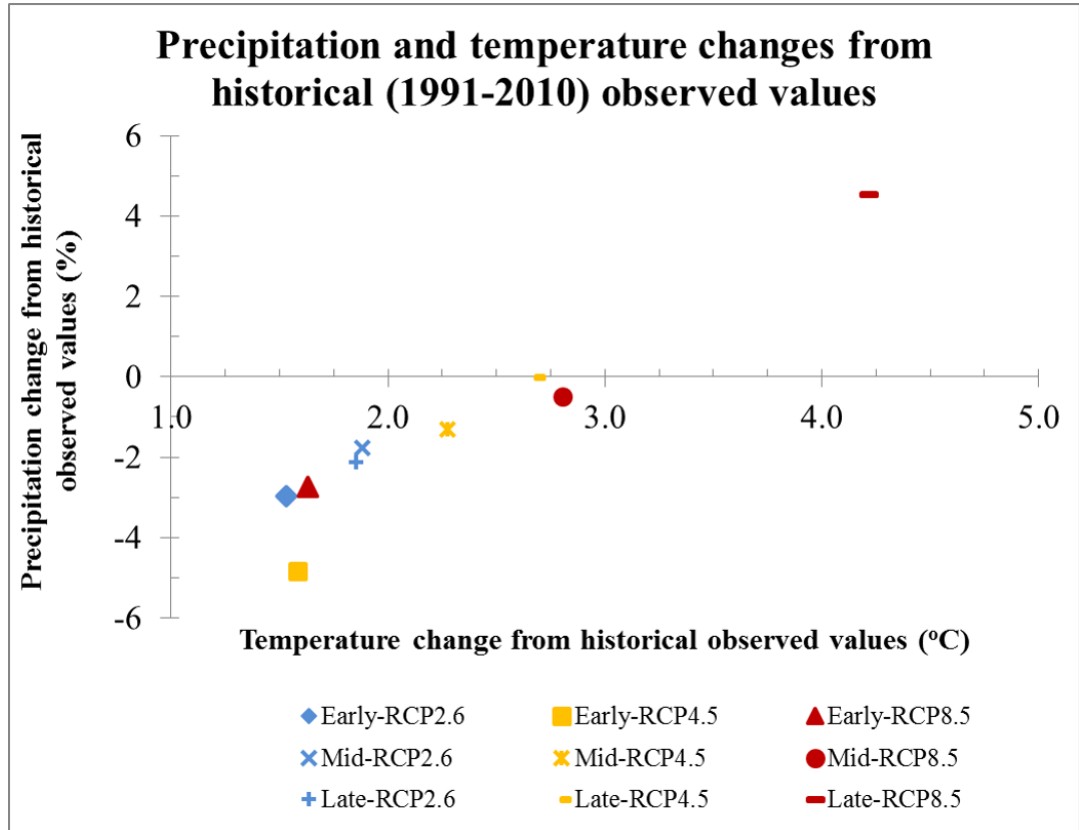

Figure 3: Average annual precipitation and temperature changes for the three RCP scenarios in early, mid and late century compared to historical (1991-2010) observed values.





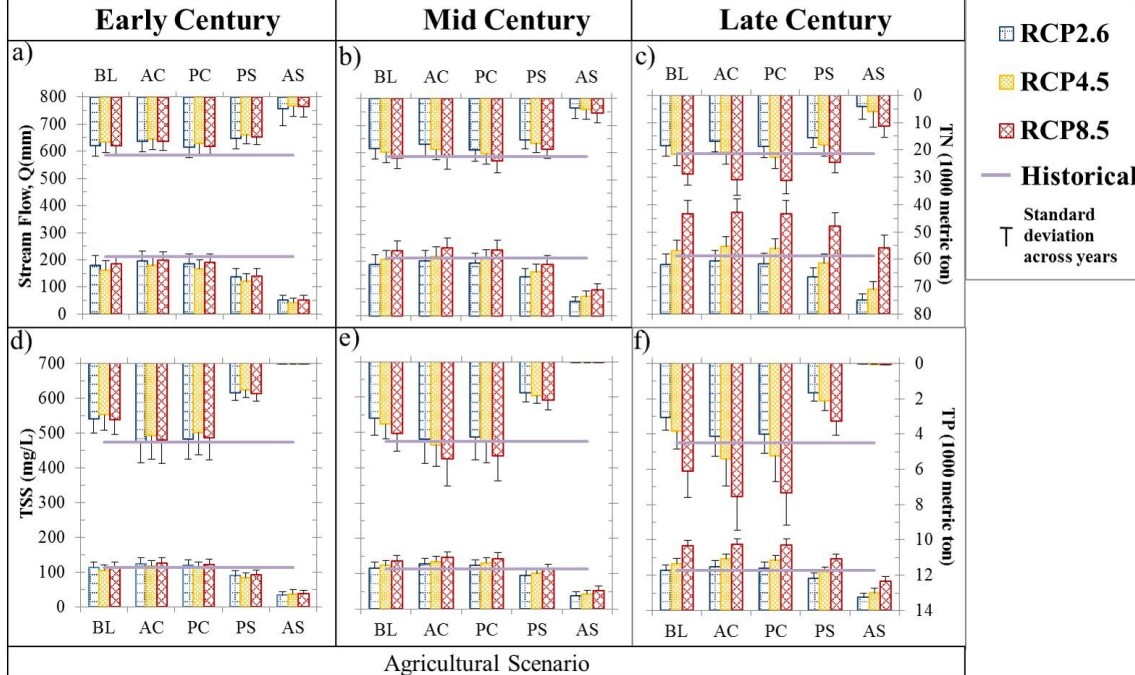

Figure 4: Stream flow (Q), total suspended solid (TSS), total nitrogen (TN) and total phosphorous (TP) results at the outlet of the watershed in different agricultural and climate scenarios





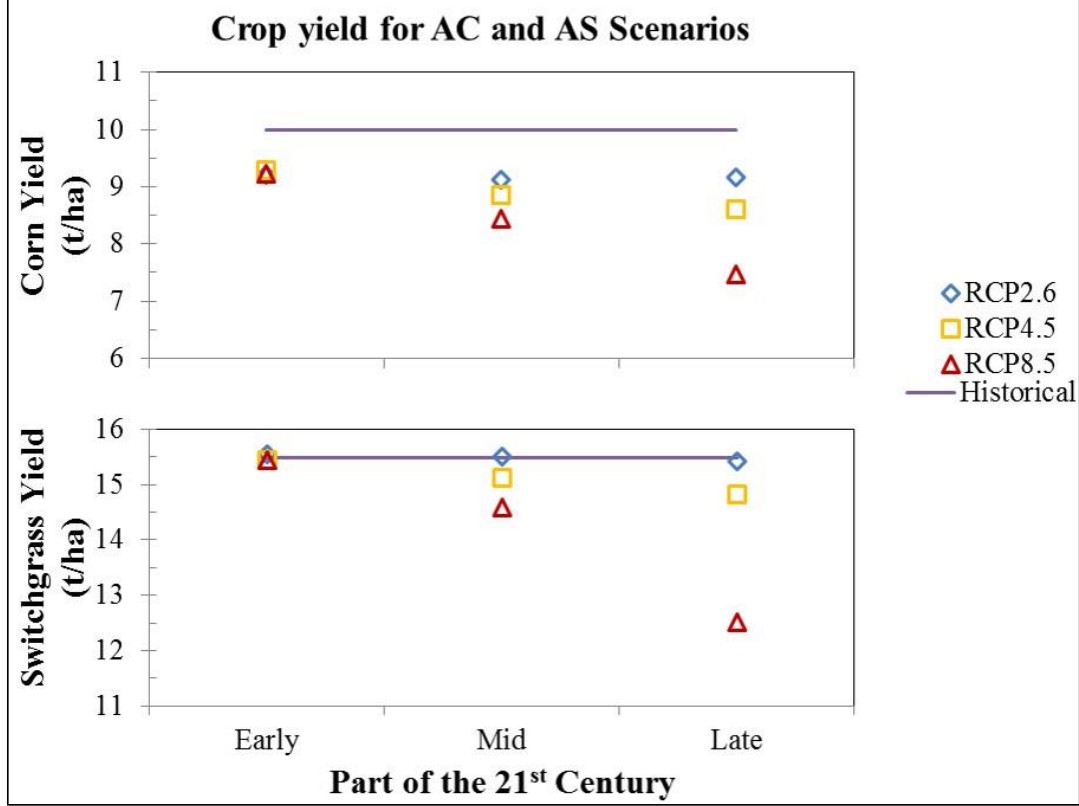

Figure 5: Watershed average crop yield for corn and switchgrass using AC and AS agricultural scenarios, respectively.