# Peer review of "Assessment of impacts of agricultural and climate change scenarios on"

_Hydrology and Earth System Sciences, 2016_

## Referee Comment (RC1) · Anonymous Referee #1 · 4 Apr 2016

General comments: The study used a distributed hydrological model (SWAT) to predict the impacts of future climate and land use changes on stream flow, total suspended sediment, total phosphorous, total nitrogen, and crop yields in a small basin in US. The authors used five agricultural scenarios and 72 climate scenarios (eight climate models*three emission scenarios*three temporal scenarios) to drive a well-calibrated model. The analysis was then based on these virtual experiments. In general, this is a typical model-based study about predicting the future changes in water quantity and quality. The data and methods used in the study were reasonable, and I think the predicted results were also reasonable and may be a reference for decision-making. Moreover, the manuscript was well organized and written. Thus, I have no major comments about this manuscript.

Specific comments: 1. Why were simulated total suspended solids different for different agricultural scenarios? I think the mechanisms about the TSS simulations in the model should be introduced briefly in section 3.1. 2. I think the major contribution of this study is that the authors analyzed of the combined effects of agricultural land use change and climate change. However, I found that the scientific questions are lacking. Can the agricultural scenarios be completely independent with the climate scenarios? Is it necessary to consider the adaption of agriculture to climate change? Moreover, the basin is too small. Are the conclusion representative for the whole U.S. Corn Belt region?

Technical corrections: P, L7: "Addition research" should be "addition research"

---

## Referee Comment (RC2) · Anonymous Referee #2 · 18 Apr 2016

This is a very useful manuscript in the context of bioenergy production within a hydrologic modeling framework. Even though several manuscript published in similar direction, I feel this manuscript is informative for research community.

I have following comments:

1. Provide a brief description on SWAT model calibration/validation with respect to water quantity/quality.

2. Considering the fact that the climate, land use and crop pattern will change in future, how did you deal model parameter uncertainty? Are you considering the historical model parameters for future scenarios as well?

[Figure]

3. Is there a role of groundwater contribution to hydrologic modeling?

4. Please be specific what major conclusions were derived from the study?

---

## Referee Comment (RC3) · Anonymous Referee #2 · 24 Apr 2016

Authors have addressed my comments. Manuscript can be accepted.

---

## Referee Comment (RC4) · M. Ashok (Referee) · 24 Apr 2016

Authors have addressed my comments. Manuscript can be accepted.

---

## Author Response (AR1)

**Responses for Referees' Comments**

**Referee #1**

Q1: Why were simulated total suspended solids different for different agricultural scenarios? I think the mechanisms about the TSS simulations in the model should be introduced briefly in section 3.1.

Answer 1: The same could be said for flow, total nitrogen and total phosphorous yields. The theoretical manual for the model used in this study (SWAT) clearly describes processes affecting and equations used in determining water, sediment, nitrogen and phosphorous yields, in addition to other parameters. Therefore, in the manuscript we invited the reader to refer to the manual and additional literature (P6, L11) for description of each components of the model. The focus of this study is using the model for long term impact analysis, not evaluating certain components of it

Q2: I think the major contribution of this study is that the authors analyzed of the combined effects of agricultural land use change and climate change. However, I found that the scientific questions are lacking. Can the agricultural scenarios be completely independent with the climate scenarios? Is it necessary to consider the adaption of agriculture to climate change?

Answer 2: We appreciate the reviewer for these insights. As described in P9, L3-23, land use change (agricultural or others) is one component embedded in climate models to determine the possible GHG concentration pathways or RCPs. While we describe what each RCP represents in P9, L3-14, the interdependence between RCPs and agricultural scenarios was briefly described in P9, L15-23. In the results section, P12, L19-25, we offer a way to look at the results considering this interdependence between climate and agricultural scenarios. We will expand this section in our revision.

Q3: Moreover, the basin is too small. Is the conclusion representative for the whole U.S. Corn Belt Region?

Answer 3: The Raccoon River watershed (RRW) is a typical Corn Belt Region (CBR) watershed with its intensively tiled fields dominated by annual crop farms. That is why CBR is used throughout the manuscript. The value of the paper is partly in the high spatial resolution of the analysis, which could help guide conservation policy in the future. To provide some context, in the US, conservation planning is generally done at a HUC 12 level, and there are 108 HUC 12 subwatersheds in the RRW, so in that sense the watershed is quite large. Larger scale studies use coarser data, and therefore are generally not suitable for immediate use in fine-grained conservation use. To provide further context, there are not many watersheds in the CBR with water quality data as good as the RRW. The water quality data provide a really sound basis for calibration and validation of the model, which is not always possible in watersheds where, for example, only flow data is collected. Thus, the results in this study can be used as a reference or starting point for future studies in similar watersheds, especially in the CBR. However, we do not claim our study for the RRW to represent the entire CBR. There need to be similar studies for other watersheds in the region, and larger scale studies for the entirety of the region to derive reliable conclusions and recommendation for the whole region and its impacts on downstream water quality.

**Referee #2**

Q1: Provide a brief description on SWAT model calibration/validation with respect to water quantity/quality.

Answer 1. We published the entire calibration/validation of the SWAT model for water quantity and quality in a previous open access paper: http://link.springer.com/article/10.1007/s00267-015-0636-4, DOI: 10.1007/s00267- 015-0636-4 (Teshager et al. 2015). Since the calibration/validation process is quite extensive, and linked to a novel method to categorize land use and build HRUs, we referred readers to look at that publication for any information on calibration/validation of the SWAT model used in this manuscript. In this specific manuscript, however, essential information

about the SWAT model and water quantity/quality calibration/validation procedures is discussed in Sect. 3.1.

Q2. Considering the fact that the climate, land use and crop pattern will change in future, how did you deal model parameter uncertainty? Are you considering the historical model parameters for future scenarios as well?

Answer 2. We assume that the historical parameters will stay the same in our future scenarios. This is an excellent point, and the reviewer's insight on this issue is very welcome. For example, crop production technology may change in the future which may change historical crop parameters used in the model. Climate change may also have an impact in changing some of the historical crop parameters. These changes in turn will affect water quantity/quality yields. Hence, we will make sure to incorporate this comment and point out the issue in our conclusion section in the revised manuscript.

Q3. Is there a role of groundwater contribution to hydrologic modeling?

Answer 3. Yes. In SWAT hydrological modeling, groundwater is one of the hydrological components that contribute to total water quantity and quality yields. Total water yield in SWAT is the summation of surface flow, lateral flow, tile flow (if applicable, - in our area there is substantial tile flow) and groundwater flow in excess of pond abstraction and transmission loss. Total nutrient yields are therefore dependent on nutrient contributions from each of the components listed above. Detailed descriptions can be found in the SWAT input/output and theoretical documentations (http://swat.tamu.edu/documentation/). In this specific study, explicit discussion about the groundwater component is not part of the objective. As a result, groundwater contributions were not specifically discussed. In a previous study (http://link.springer.com/article/10.1007/s00267-015-0636-4), however, we have discussed the importance of groundwater flow contribution in the watershed in terms of baseflow. Hence, we encourage readers to refer the respective paper and SWAT manuals mentioned above for more information.

Q4. Please be specific what major conclusions were derived from the study?

Answer 4. We thank the reviewer for this comment. We believe that one important conclusion of our work that could be clarified in the conclusion part of the manuscript is that there are significant trade-offs in protecting water quality in intensive agricultural regions that could be exacerbated by climate change, for example, planting more switchgrass would benefit water quality but negatively impact food production. However, there is also potential for win-win situations – if biofuels from switchgrass become commercially viable, that will reduce the pressure on corn. Another conclusion we could have better outlined is that, given climate change impacts, our results suggest that substantially improving water quality will require a combination of working land practices (such as conservation tillage and cover crops) and land retirement/perennial plantings (such as planting grasses such as switchgrass). This will require substantive conservation efforts, higher than historical levels. We will expand on this issue in the conclusion part of our manuscript.

**Relevant Changes Following Referees' Comments**

1. Following "referee #1 Q2" comments, we have expanded the "Conclusion" section to clarify more on the interdependence between climate scenarios and agricultural scenarios we discussed in our "Simulation" and "Results and Discussion" sections. The following were added and/or modified:
    a. P17, L15-28:
    "It is also important to consider how the agricultural scenarios modeled for the RRW fit with the forcing scenarios, and with the larger context of agricultural adaptation to climate change at a global scale. Specifically, the AS and PS scenarios would be compatible with the RCP2.6 pathway if coupled with sustainable intensification of agricultural practices and advanced biofuel production (Melillo et al., 2009; Tilman et al., 2011; Foley et al., 2011). Otherwise, the reduction in corn production from areas such as the RRW would result in more environmental degradation, deforestation and higher carbon emissions elsewhere. Conversely, it is possible – though not likely – that the AS and PS scenarios could occur in a high emission world, if strong conservation measures were to be limited to the US. Similarly, the AC scenario might be compatible with the low emission RCP2.6 pathway if effective conservation measures to reduce deforestation were implemented at a global scale, but US conservation policies lagged behind. This illustrates the importance of the interplay of national and global conservation policies in addressing the challenge of climate change. In general, in order to promote local water quality in heavily farmed watersheds such as the RRW, as well as reducing global GHG emissions, more complex landscapes and serious conservation measures will have to be put into practice across the planet."
    b. P18, L6-7:
    "This is even more important if we consider how likely it is that agriculture will likely develop technologies to adapt to climate change."

2. Following "referee #2 Q2" comments, we have pointed out the issues related to parameter uncertainty with respect to using historical parameters for future scenarios in our "Conclusion" section. The following were added and/or modified:
    a. P18, L1-6:
    "We should also point out that model parameters used during calibration and validation periods were kept the same for our future scenario simulations. This assumption could carry more model parameter uncertainties in scenario simulations depending on the extent of future technological and climate changes. For example, in the last century there have been large changes to the technologies used in agriculture – from synthetic fertilizers to new hybrids to precision agriculture. If such considerable changes were to continue, the impacts on water quality could be significant."
    b. P18, L7-8:
    "Hence, future studies should devise a way to take these potential effects into account when parametrizing SWAT modeling for future scenario analysis"

3. Following "referee #2 Q4" comments, we have modified and expanded our "Conclusion" section to articulate the major conclusions drawn from our study. The following were added and/or modified:
    a. P17, L2-5:

[revised manuscript text omitted]